# Design of High-Power Solid-State Transformers with Grain-Oriented Electrical Steel Cores

Daniel Roger [1,*] , Ewa Napieralska [1] , Krzysztof Komeza [2] and Piotr Napieralski [3]

1 Laboratoire Systèmes Electrotechniques et Environnement (LSEE), Université d'Artois ER4025, 9 Rue du Temple, 62000 Arras, France; ewa.napieralskajuszczak@univ-artois.fr
2 Institute of Mechatronics and Information Systems, Lodz University of Technology, Stefanowskiego 18/22, 90-001 Lodz, Poland; krzysztof.komeza@p.lodz.pl
3 Institute of Information Technology, Lodz University of Technology, Wólczańska 215, 90-924 Lodz, Poland; piotr.napieralski@p.lodz.pl
* Correspondence: daniel.roger@univ-artois.fr

**Abstract:** The paper proposes a simple structure of high-power solid-state transformers (SSTs) able to control the energy flow in critical lines of the medium-voltage (20 kV) distribution grid. With an increasing number of renewable intermittent sources connected at the nodes of the meshed distribution grid and a reduced number of nodes connected to large power plants, the distribution grid stability is more and more difficult to achieve. Control of the energy flow in critical lines can improve the stability of the distribution grid. This control can be provided by the proposed high-power SSTs operating a 20 kV with powers over 10 MW. This function is difficult to achieve with standard SST technologies that operate at high frequencies. These devices are made with expensive magnetic materials (amorphous or nanocrystalline cores) and a limited power by SST cells. The required total power is reached by assembling many SST cells. On the other hand, existing SST designs are mainly aimed at reducing the equipment's size and it is difficult to design small objects able to operate at high voltages. The authors propose to use cores made with grain-oriented electrical steel (GOES) thin strips assembled in wound cores. Experimental results obtained, with GOES wound cores, show that the core losses are lower for a square voltage than for a sine one. This counterintuitive result is explained with an analytical calculus of eddy currents and confirmed by a non-linear time-stepping simulation. Therefore, simple converter structures, operating with rectangular voltages and low switching losses, are the best solutions. Experimental results also show that the core losses decrease with temperature. Consequently, high-power SST cells can be made with transformers whose GOES cores are hotter than coils for reducing core losses and keeping copper losses at low levels. The paper proposes an appropriate transformer mechanical structure that avoids any contact between the hot GOES wound core and the winding, with a specific cooling system and thermal insulation of the hot GOES wound core. The proposed design makes it possible to build SST cells over 1MW and full SSTs over 10 MW at moderate costs.

**Keywords:** solid-state transformer (SST); grain-oriented electrical steel (GOES); core losses; wound core; eddy current losses; high-power; Litz wire

## 1. Introduction

Nowadays, the reduction of greenhouse gas emissions is a major challenge for society to limit the planet's global warming, [1–3]. To reduce the part played by coal, oil and gas in the production system with a fixed number of nuclear power plants, the production of renewable energy must sharply increase within a few years [4] and the electric grid must adapt [5–7]. The grid connects the power sources to the end-users by a complex mesh of electrical lines and transformers. The grid must ensure electric power availability, despite local faults. At a large scale, electricity storage is very expensive; therefore, at any time, the total power of the sources must be equal to the total power consumed by

end-users. If it is not the case, the frequency increase or decreases with a slope that depends on the global inertia of the production synchronous machines. The global demand of end-users is determined by complex statistic tools that impose the global production at any time. However, the power of the renewable sources, which depends on weather, is not correlated to the global demand. The difference must be compensated by gas or hydraulic turbines which are rapidly controllable sources [8]. Nowadays, the part of the rapidly controllable sources remains low compared to the total power of the large power plants. In the lines of the grid, the power is mainly in one direction: toward the end-users. With more intermittent and distributed sources based on wind or sun and fewer gas turbines, the grid stability is more difficult to achieve [9,10]. Local energy storage devices will be necessary for obtaining a stable frequency [11]. Power direction may change in several lines according to the intermittent local production. The compensation of the voltage drops in these lines is much more difficult to perform; it needs to act on the energy flow in both directions, in critical lines of the grid [12]. This function can be achieved by on-load tap changers [13], whose principle is to modify the number of turns of power transformer coils by a moving mechanical contact inside the transformer oil [14]. Such systems, with sliding contacts in on-line power transformers, produce sparks in the contact area at each movement. These sparks create mechanical wears and oil pollutions. A regular maintenance is necessary; it can now be guided by sophisticated fault detection systems [15]. High-power SSTs can perform all the functions of on-load tap changers (galvanic insulation, voltage change and voltage tuning) much faster and without any specific maintenance demand. The concept of SST in born in the 70′s with the availability of power electronic switches [16] and developed later [17]. A synthetic review paper explains the origin and the evolution of SST's key concepts [18]. Several more recent papers give the principles of SST applications in distribution systems [19–23]. High-power SSTs are also much smaller than standard transformers for the same powers; therefore, they are also suitable for railway applications [24,25].

In SSTs, the galvanic insulation is ensured by transformers. Their sizes are all the smaller as the operating frequency is high. The limits are linked to power losses in electronic components, in cores and in the windings. Global thermal management and economic considerations must also be considered. Transformers made with ferrite cores and Litz wires have existed for a long time, they can operate at very high frequencies (over 20 kHz) but at peak flux densities under 0.4 T [26,27]. The ferrite type selection is an important part of the SST design procedure [28]. For such frequencies, the power is limited to a few tens of kW [29]. High-power can be obtained by assembling many cells, with a high global cost for the whole SST. Cells of higher powers can be designed with amorphous or nanocrystalline magnetic cores that can operate at higher flux densities (1.2 T) but lower frequencies [30,31]. These materials have relative low Curie temperatures. If the maximum temperature defined by the manufacturer is exceeded during an accidental overload, the core can permanently have lower performances [32,33]. They are also more expensive than ferrites. Papers [34–36] propose three key elements: a method helping to choose the best magnetic material; they propose a design and an implementation of a high-power high-frequency transformer and cooling pipes around the core in order to reduce the hot spots.

Amorphous and nanocrystalline soft cores can operate up to 150 °C [37]; the maximum operating temperature of ferrites is slightly lower [26]. These temperatures are under the temperature class of standard enameled wires used for windings (class H: 180 °C, class C: 200 °C). Consequently, the thermal limit of the whole transformer is imposed by the core. The electrical insulation systems (EISs) is not used at its rated characteristics for both core technologies. Grain oriented electrical steel (GOES) can also be used for building SST transformers. This core technology is suitable for slightly lower frequencies and higher peak flux densities. Papers [38,39] present a 30 kVA–3 kHz single-phase transformer and a 200 kVA three-phase one operating at 1 kHz with GOES cores. The material crystalline structure is very stable; the material takes the benefits from a long industrial experience that demonstrates a very long-life expectancy [40]. The GOES has a Curie temperature

much higher than other materials; it can be used up to 500 °C [41]. GOES in thin strips are cheaper than amorphous and nanocrystalline soft magnetic materials. Therefore, GOES cores can be used for designing high-power SST cells at a reasonable cost.

The paper proposes a new concept for designing medium-frequency high-power transformers (in the MVA range) for SST cells. The proposed concept uses the good properties of GOES thin strips at high temperatures. With these transformers and standard 6.5 kV insulated gate bipolar transistors (IGBTs) modules [42], it is possible to design very large power SSTs (several MW) at a reasonable cost, with a limited number of cells [43]. The central transformer of each cell uses a wound core made of thin GOES strips. In such cores, the magnetic field is in the strip rolling direction at any point, which corresponds to the easy magnetization axis. This direction corresponds to the higher magnetic permeability and saturation flux density. The high permeability allows a core design with long strips that offer a large core window for the winding. The first part of the paper describes the structure of a high-power SST and of an elementary cell as simple as possible. The following section presents core losses measurements made on a wound GOES core for sine and square voltages. These losses are lower for square voltages than for sine ones. A numerical simulation and an analytical analysis of eddy current losses explain these counter-intuitive experimental results. This section also shows that the core losses are lower at high temperatures than at usual ones. The end of the paper gives key elements for designing a high-power medium-frequency transformer with a core at a much higher temperature than for coils in order to increase the transformer compacity.

## 2. High-Power Solid-State Transformer (SST)

### 2.1. General Structure

High powers can be reached by associating several SST cells. Figure 1 gives an example of modular SST used to connect a high-voltage DC bus to a low-voltage one. This figure shows also that the galvanic insulation stress of the upper transformer (cell N) of the whole high-voltage DC bus, which is a key point of the transformer design.

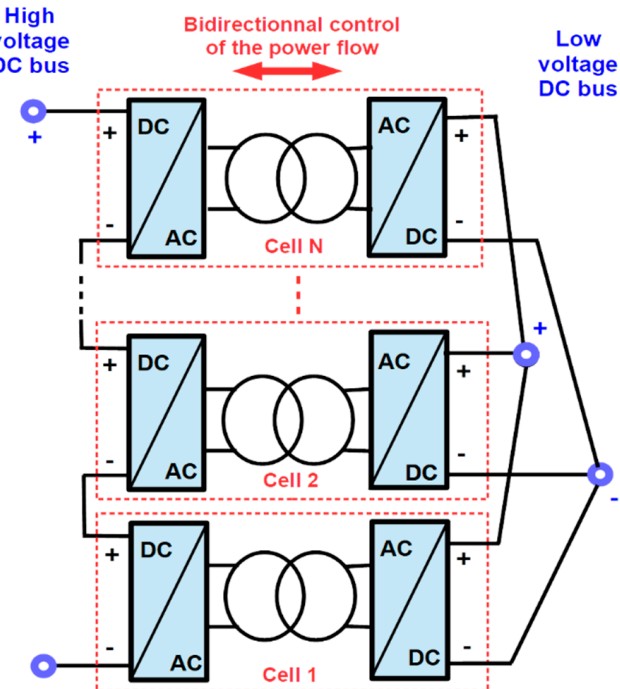

**Figure 1.** Modular structure of the DC/DC heart of a high-power SST connecting a high voltage grid to a low voltage one. The SST cells input connected in series allows operating with a high-voltage DC bus while a high output current is obtained by coupling the cells output in parallel.

When the problem is to connect an 20 kV AC grid to a 400 V one, the DC/DC SST is placed between two AC/DC reversible thyristor bridges. For this example, the high-voltage DC bus is 20 $\sqrt{2}$ = 28.2 kV and the low-voltage one slightly over 400 $\sqrt{2}$ = 566 V. The choice is to set the low-voltage DC bus at 650 V for having a tuning capability of 10% over the rated value. The heart of the SST can be composed of 5 cells made with 6.5 kV/200 A IGBT legs, which corresponds to a total power of 7 MW (1.4 MW per cell). The low voltage converter is made of several standard low-voltage IGBT legs connected in parallel for operating at high currents.

Each SST cell key element is a transformer placed between two electronic bridges. This structure, called "dual active bridge (DAB)" [44,45], is presented in Figure 2. The filtering capacitances $C_A$ and $C_B$ must be large enough to keep the input and output voltages $U_A$ and $U_B$ constants over a short timescale (a period), while these voltages can change over a longer timescale that corresponds to the grid voltage tunings.

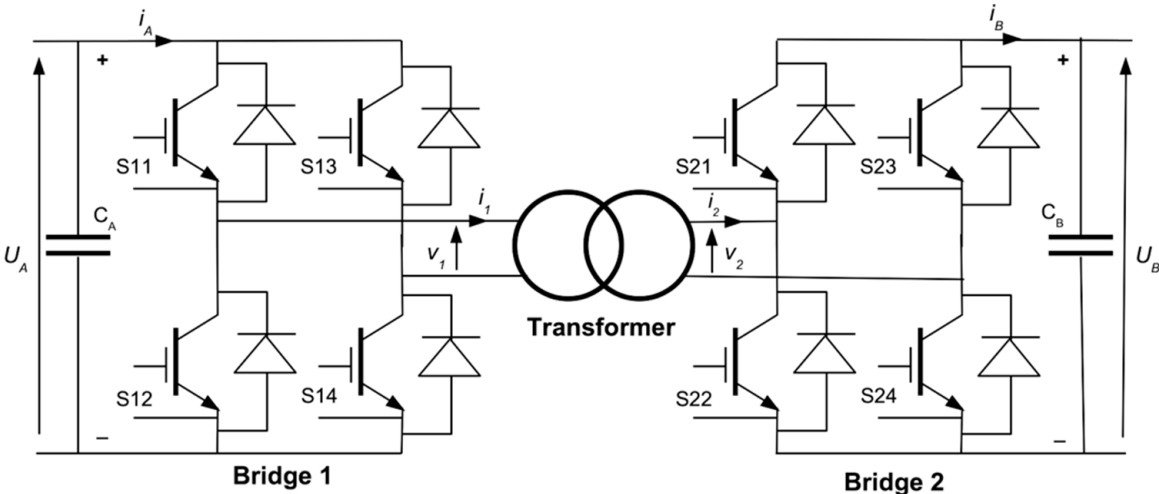

**Figure 2.** Structure of a dual active bridge (DAB) with its central transformer providing the galvanic insulation and voltage change. The power control can be made by imposing the phase lag between bridge 1 and bridge 2.

The losses in the first electronic bridge (S11 … S14) are the sum of the conduction $P_{cond}$ and switching losses $P_{sw}$. Noting $V_{CEsat}$ the saturation voltage of an IGBT, $t_r$ and $t_f$ the rise and fall times of the collector current in an IGBT, $U_A$ the DC input voltage, $<i_A>$ the mean value of the input current, and $f_s$ the switching frequency, the losses in an IGBT bridge are given by Equations (1)–(3).

$$P_{bidge1} = P_{cond1} + P_{sw1} \tag{1}$$

$$P_{cond1} \approx 2V_{CEsat} < i_A > \tag{2}$$

$$P_{sw1} \approx U_A < i_A > \left( t_r + t_f \right) f_S \tag{3}$$

Assuming that $V_{CEsat}$ is constant, Equation (2) shows that the conduction losses depend only on the average current $< i_A >$; Equation (3) shows that the switching losses depend on $U_A < i_A >$, which is the input power, and on the switching frequency $f_S$. For a given input power, the global losses (1) are all the weaker as the voltage is high and the switching frequency $f_S$ low. Expressions (1–3) are similar for bridge 2; the same reasoning can be applied with an imposed voltage $U_B$.

Supposing a constant input and output voltages ($U_A$ , $U_B$ ) over a short timescale, the power at the input Bridge 1 is $P_A = U_A < i_A >$ and $P_B = U_B < i_B >$ at the output of the bridge 2. Consequently, it is important to avoid pulse width modulation (PWM) commands for operating at the lowest possible switching frequency. This command corresponds to the

rectangular voltages presented in Figure 3. The control variable is $\varphi$, the phase lag between the commands of the two IGBT bridges.

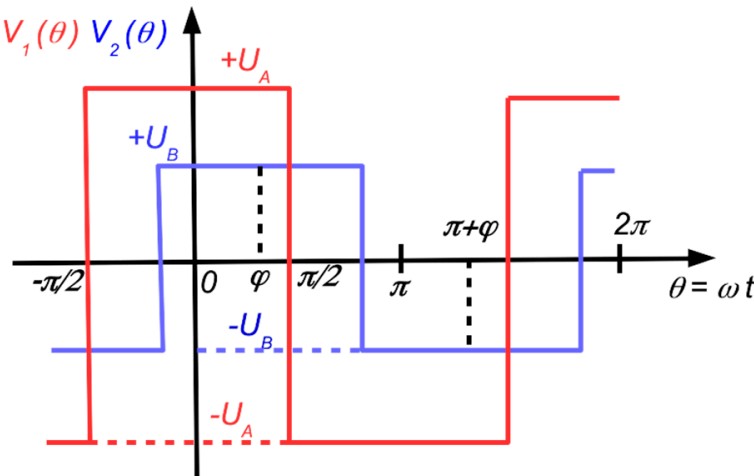

**Figure 3.** Rectangular voltages are imposed on the primary and the secondary coils of the transformer by the two IGBT bridges; $\varphi$ is the control variable.

Noting $n_1$ and $n_2$ the turn number of the primary and the secondary coils, $\lambda$ the transformer leakage inductance seen from the primary and neglecting the coil resistances, the transferred power between the input and the output is given by Equation (4).

$$P = \frac{U_A \, U_B \, (n_1/n_2)}{\pi \lambda \omega} \; \varphi(\pi - |\varphi|) - P_{bidge1} - P_{bidge2} \tag{4}$$

### 2.2. Medium-Frequency Transformer Made with a GOES Wound Core

Figure 4 shows an example of a core made of two identical half-cores. Each half-core is built by winding thin GOES strips on the support that imposes the shape of the window. After the thermal annealing cycles, the support is removed, and the half-core is cut to facilitate the coil insertion. The cutting planes are machined to obtain very flat surfaces and a minimum magnetic gap.

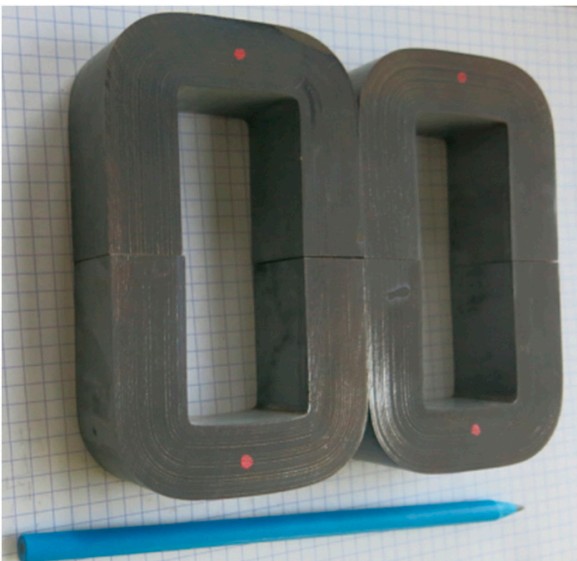

**Figure 4.** Example of wound core made with thin GOES strips insulated by a ceramic coating.

Each GOES strip is covered with thin ceramic layers $(3 - 5 \; \mu m)$ on both faces. These inorganic layers provide an electrical insulation able to operate at very high temperatures.

On the other hand, the crystalline structure of this material is very stable. Consequently, GOES wound core can be used up to 500 °C with only a small reduction of the maximum flux density [41]. The tests are performed with a wound core made with a strip whose effective thickness is $a = 165$ μm and core width $b = 25$ mm. The average length of the wound core is $l = 330$ mm.

Core loss measurements are made at medium frequencies with a testing bench shown in Figure 5, for square and quasi-sine voltages. Two identical 10-turn coils are placed around the central leg of the magnetic core. The primary coil is fed directly by the inverter for measurements with square voltages. The low-pass filter, tuned at a frequency slightly under the inverter one, strongly reduces the harmonics and provides à quasi-sine voltage. The inverter voltage is increased to compensate for the filter voltage drop at the measurement frequency. The second-order filter reduces the 3rd harmonic 9 times compared to the fundamental one and more for upper harmonics. The harmonic distortion ratio of the quasi-sine voltage is 3.8%.

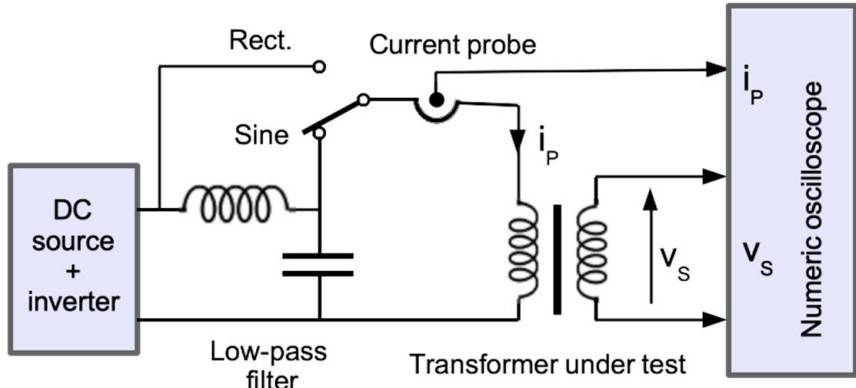

**Figure 5.** Testing bench for core loss measurements with square and quasi-sine voltages at medium frequencies. Data are recorded by the numeric oscilloscope on a time window that corresponds to an exact period number and with 100,000 points.

A current broadband probe gives the magnetizing current while a voltage probe measures the secondary voltage. A numeric integration of the secondary voltage yields the average flux density in the core at any time. The core losses are computed using the primary current and secondary voltage to avoid the influence of copper losses in the primary coil. The computation applies the basic definition of the power with is the average value, on an exact number on periods, of the products of the recorded instantaneous values. Figure 6 compares the specific losses in the GOES wound core presented in Figure 4 at 2 kHz for square and quasi-sine voltages. For each measurement point, the inverter voltage increases rapidly from zero to get the desired peak flux density; data are recorded; the voltage is quickly reduced to zero. A deadtime of several minutes between consecutive measurements avoids a significant increase in the core temperature during tests. Figure 6 shows the classical increase in losses with the flux density. However, the losses remain lower for square voltages than for sine ones for the same peak flux density. Similar results are obtained for other frequencies over 1 kHz.

The constatation of lower core losses for square voltages rather than for quasi-sine ones is counterintuitive because, for many loads, the harmonics of a square voltage create additional losses. A detailed theoretical explanation is proposed considering eddy currents in GOES strips over 1 kHz. Experimental results obtained with the same testing bench also show that the global core losses decrease when the core temperature increases, as shown in Figure 7 for two operating points 2 kHz–1.3 T and 3 kHz–0.8 T. This reduction is due to the natural increase in the metal resistivity with temperature.

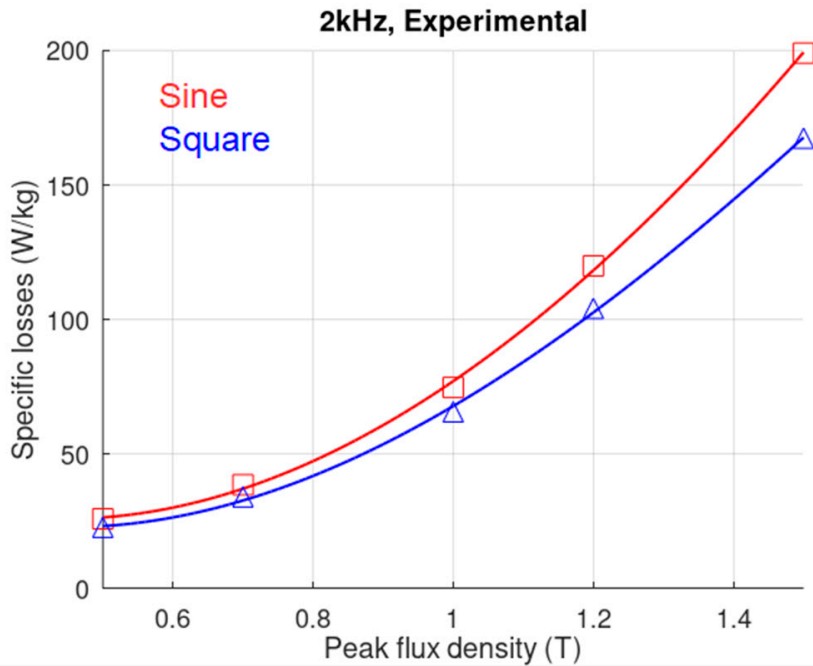

**Figure 6.** Specific losses in a GOES wound core for sine and square voltages.

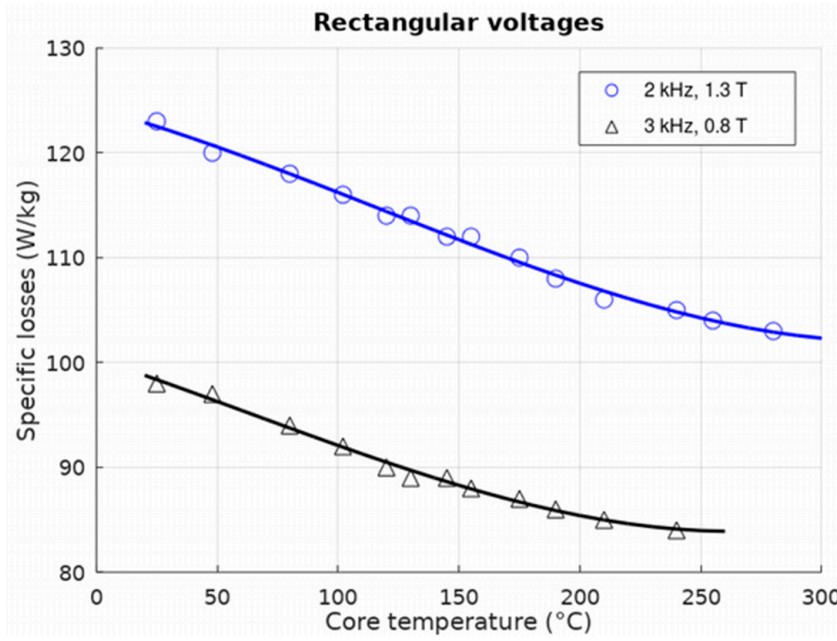

**Figure 7.** Core loss variations with the core temperature for rectangular voltages and for two operating points 2 kHz–1.3 T and 3 kHz–0.8 T (experimental results).

### 2.3. Interpretation Using Non-Linear Time Stepping Simulations of Eddy Current Losses

The simulation is made with GetDP [46] considering a small part of a single GOES strip. The square voltage imposed on the primary coil corresponds to a triangular flux in the core. The 2D software is based on a standard potential vector formulation. With this formulation, it is possible to impose a flux in the strip by fixing the potential vector at the boundaries (gray lines in Figure 8). Therefore, the magnetic flux density $\vec{B}(x,t)$ is in the $y$-direction, and the current density $\vec{J}(x,t)$ in the magnetic material is in the $z$-direction. Both quantities depend on $x$ because of the skin effect. The instantaneous value of eddy

current losses $p(t)$ is computed by Equation (5), where $N_S$ is the number of strips in the wound core, $l$ their average length, and $\sigma$ the material conductivity.

$$p(t) = N_S \, 2 \, l \, b \int_{-\frac{a}{2}}^{\frac{a}{2}} \frac{1}{\sigma} \, J^2(x,t) \, dx \tag{5}$$

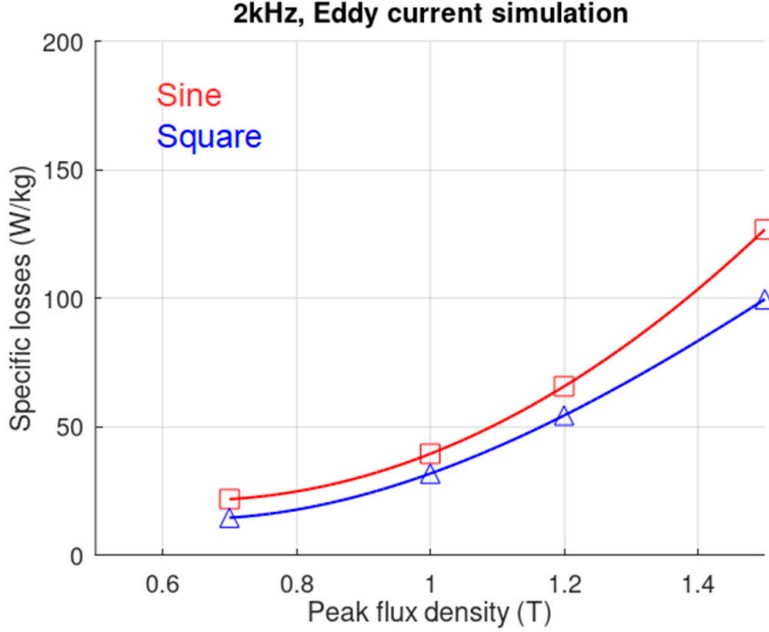

**Figure 8.** Small part of a strip of thickness $a$ and width $b$ with the associated cartesian reference. The flux density is in the rolling direction $y$.

The time-stepping simulation starts from zero; it yields a transient state before the repetitive periodic waveforms of the steady-state. The eddy current losses are computed at a steady state for an entire time period. Results are presented in Figure 9.

**Figure 9.** Eddy current losses in a GOES wound core for sine and square voltages computed by Equation (5) from the time-stepping finite element simulation results.

A comparison of Figures 6 and 9 show that eddy current losses obtained by simulation are about half the measured core losses. Bertotti's formula cannot be used for a non-uniform flux density, but the principle of losses separation remains valid. At 2 kHz, the static losses are low; therefore, the excess losses are of the same magnitude as the eddy current ones.

Figure 10 shows an example of the local flux density inside a strip at a time corresponding to the peak value of the average flux density (1.3 T) for a square voltage at 2 kHz. The skin effect is a major phenomenon, where the GOES strip is saturated near its surfaces despite a lower peak average flux density.

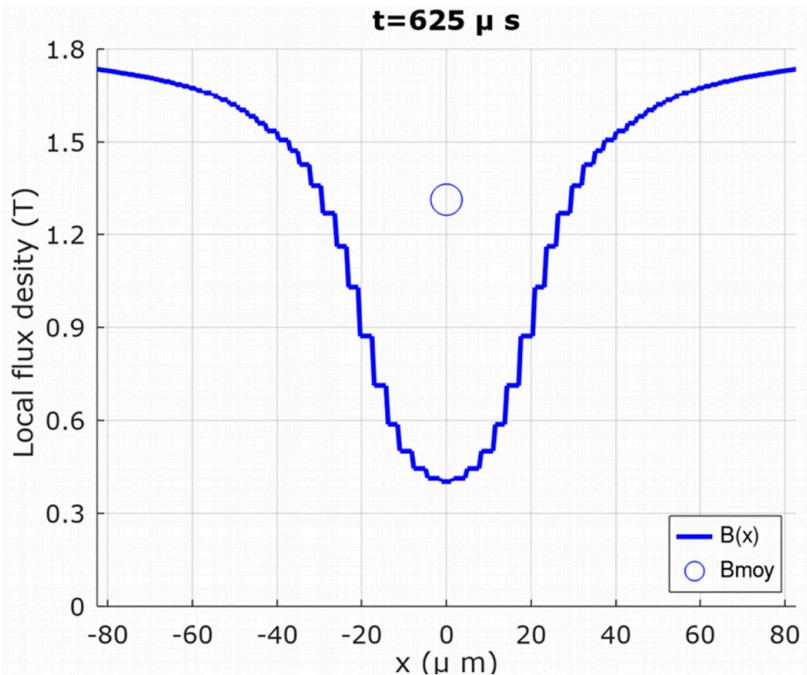

**Figure 10.** Example of a spatial distribution of the flux density inside a GOES strip for rectangular voltage at a time corresponding to the peak flux in the core.

### 2.4. Interpretation for Low Flux Densities (Linear Conditions)

For low flux densities, the magnetic permeability $\mu$ of the material can be assumed to be constant. The superposition principle required for adding the influence of harmonics can be used. The complex notation Equation (6) makes the calculus easier, where all the complex variables are underlined. The flux density $B(x,t)$ at any point inside the GOES strip is a scalar function of the position $x$ and of the time $t$ is obtained with a similar expression but with the Fourier series of the flux density $\underline{B}(x)_n$ Equation (7). This relation expresses the classical diffusion equation of a magnetic field in a conducting material. In this equation, the input variable is the Fourier series of the field at the two strip borders $\underline{H}_{0n}$.

$$H_0(t) = \sum_{n=1}^{\infty} \Re\left(\underline{H}_{0n} e^{jn\omega t}\right) \tag{6}$$

$$\underline{B}(x)_n = \mu\, \underline{H}_{0n} \frac{\cosh\left(\underline{\gamma}_n\, x\right)}{\cosh\left(\underline{\gamma}_n\, a/2\right)} \tag{7}$$

In Equation (7), $\underline{\gamma}_n = (1+j)/\delta_n$ is the propagation constant, which depends on the skin depth $\delta_n = \sqrt{2/(n\omega\mu\sigma)}$ defined for each harmonic rank $n$. The Fourier series of the current density is obtained applying Ampere's theorem, which reduces to a spatial derivation for a 1D problem.

$$\underline{J}(x)_n = -\underline{\gamma}_n \underline{H}_{0n} \frac{\sinh\left(\underline{\gamma}_n\, x\right)}{\cosh\left(\underline{\gamma}_n\, a/2\right)} \tag{8}$$

Before computing the eddy current losses with Equation (5), the current density's temporal expression Equation (9) is required.

$$J(x,t) = \Re\left(\sum_n \underline{J}(x)_n e^{jn\omega t}\right) \tag{9}$$

To apply these equations to the GOES wound core of a transformer, it is necessary to express the Fourier series $\underline{H_0}_n$ with the voltage one $\underline{v_1}_n$ that imposes the average flux density $< B >_n$ in the strip.

$$< B >_n = \frac{\underline{v_1}_n}{N_T N_S ab} \frac{1}{jn\omega} \tag{10}$$

$$\underline{H_0}_n = \frac{< B >_n a\, \underline{\gamma}_n}{2\, \mu\, \tanh\left(\underline{\gamma}_n\, a/2\right)} \tag{11}$$

Figure 11 presents the input data for sine and square voltages. The square voltage is plotted with a Fourier series cut at the rank 100. The voltage magnitudes are computed to get the same average flux density $< B >(t)$ peak values.

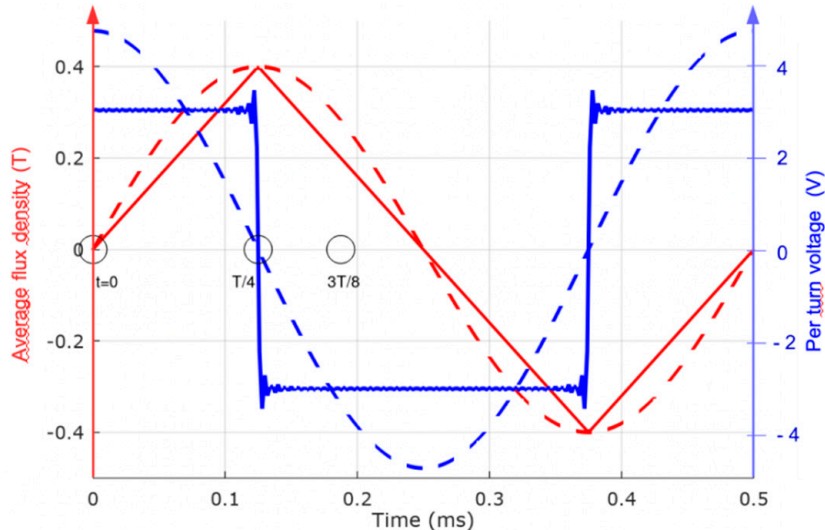

**Figure 11.** Voltage and average flux density in the core for square and sine voltages.

The flux density distributions of Figure 12 are computed using the Fourier series Equation (7) and the local power density with Equation (5) using Equation (9). The harmonics have a non-intuitive influence on the skin effect. The Fourier series must be carefully used considering the two variables $x$ and $t$, while the usual application, without skin effects, considers only one variable $t$.

Figure 13 shows that most of the power density is concentrated near the strip surfaces. However, the spatial distributions are different. At $t = 0$, for instance, the power density for a sine voltage (blue the dotted line) is much higher than for a square one (solid blue line). At $t = T/4$ (magenta lines), the situation is inversed but with fewer differences. The mean value of the power density $P_{ED}$, computed with the temporal integration Equation (9), shows that eddy current losses are lower for a square voltage than for a sine one.

$$P_{ED} = \frac{1}{T} \int_0^T p(t)\, dt \tag{12}$$

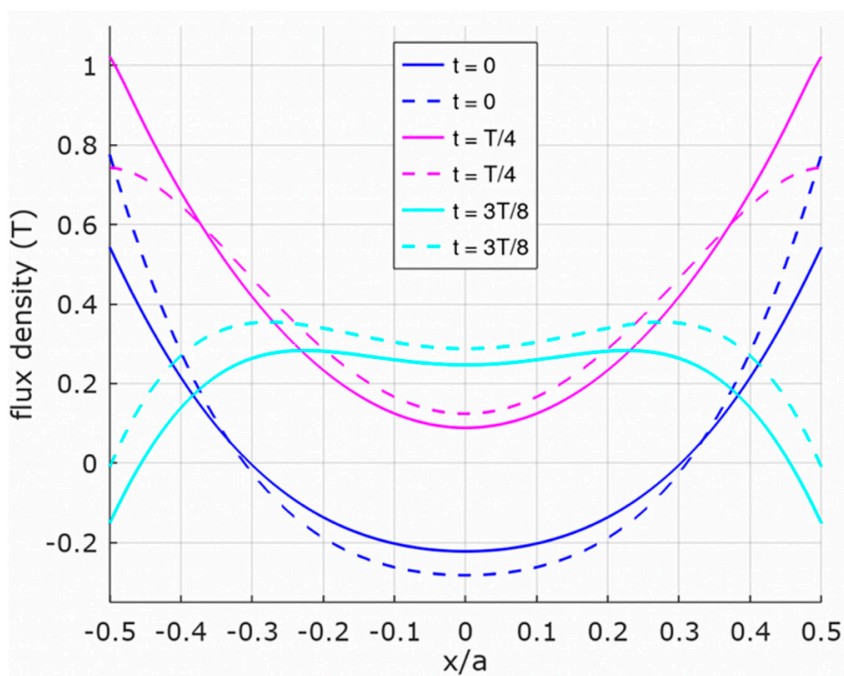

**Figure 12.** Spatial distribution of the flux densities inside a GOES strip for sine (dotted lines) and square (solid lines) voltages at several times.

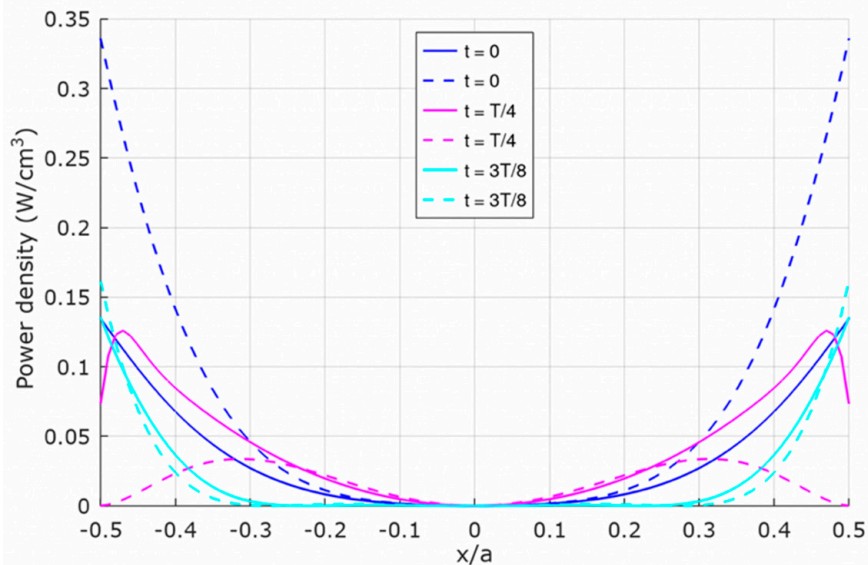

**Figure 13.** Spatial distribution of the power densities inside a GOES strip for sine (dotted lines) rectangular (solid lines) voltages at several times.

## 3. High-Power Medium-Frequency Transformer

### 3.1. Transformer Structure with a Hot GOES Wound Core

Experimental results presented in Section 2 show that the GOES wound core used at high temperatures has lower core losses than at usual temperatures. For the transformer winding, the influence of temperature is inversed because the copper resistivity increases with temperature. Consequently, a solution can be the design of a transformer with a hot core inside coils at standard temperatures for high-power applications requiring high efficiencies. This solution is possible using specific mechanical design with airflows between the core and the coils. Figure 14 presents the transformer structure with a small gap between the hot core and the winding at a standard temperature.

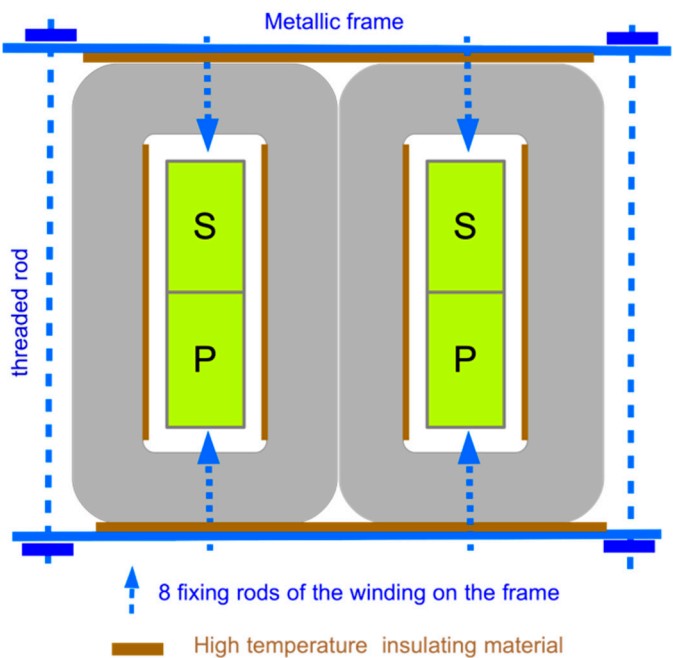

**Figure 14.** Structure of a medium-frequency transformer with a hot GOES wound core, colder coils and thermal insulation of the hot core. There is no contact point between the core and the coils.

Figure 15 details the cooling system; the top view (a) shows the free space between the core's central leg and the winding. Part (b) is a picture of one of the two small fans that provide airflows on each side of the core; it also shows two of the 8 fixing rods for the winding. There is no contact point between the hot core and the winding. With such a cooling system, the core temperature can be about 100 °C higher than that of the winding. Using a standard class H (180 °C) winding insulation technology, the core temperature is about 280 °C, which is far below the GOES capabilities (500 °C) [41]. A thermal analysis made with equivalent circuits is available in [47].

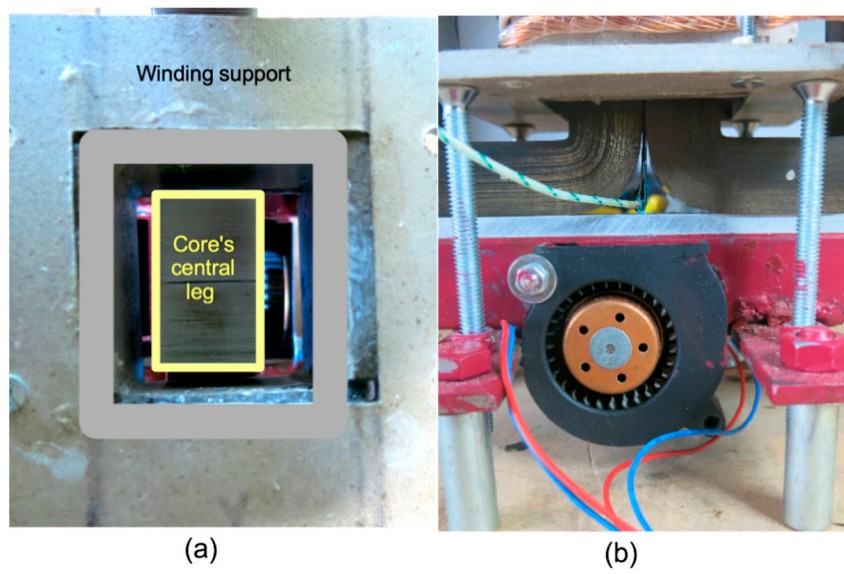

**Figure 15.** Transformer cooling system with airflows between the hot core and the winding at a standard temperature: (**a**) top view when the upper half-core is removed; (**b**) one of the two small fans providing airflows.

### 3.2. Winding Structure

Because of the skin and proximity effects, the AC resistance of coils increases with frequency. An experimental approach is proposed for considering this problem for high power medium frequency transformers, which must manage high currents with copper losses as low as possible. Three sets of two identical 16 turns are tested. Two identical coils are placed around the central leg of the core as in the picture presented in Figure 4. The core is made of ferrite to avoid core losses. The secondary is short-circuited while the impedance analyzer is connected to the primary coil with wires as short as possible. Impedance measurements are made with an Agilent 4294A precision impedance analyzer up to 10 kHz. Three wire technologies are tested: round enameled copper wire of 5 mm diameter; Litz wire made of 20 enameled copper strands of 0.7 mm diameter (Litz 1) and another Litz wire made of 84 enameled copper strands of 0.3 mm diameter (Litz 2). Experimental results in Figure 16 present the variations of the ratio $R_{AC}/R_{DC}$ giving the relative value of the AC resistance for the 3 wire technologies.

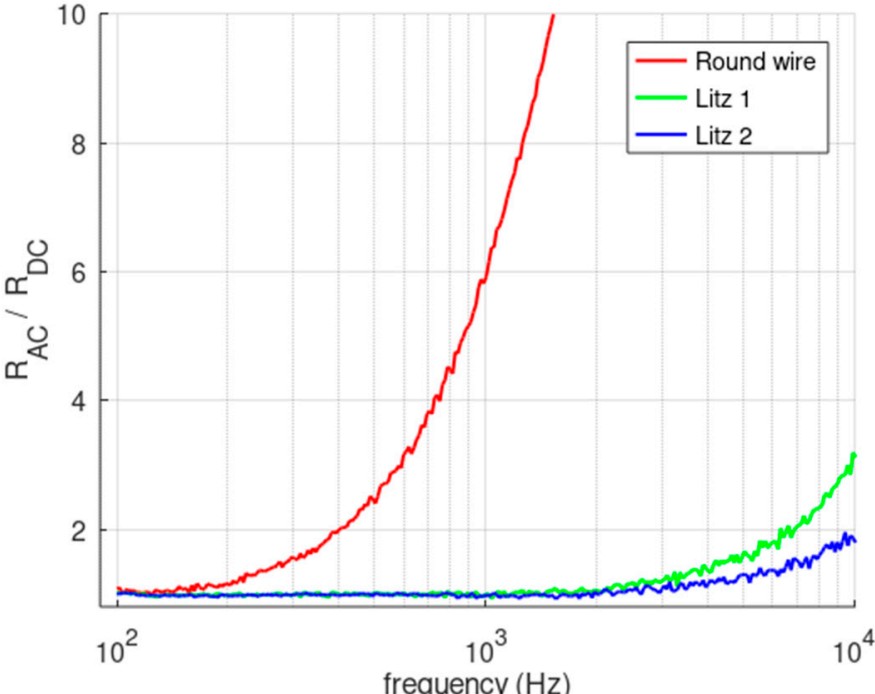

**Figure 16.** Coil resistance increase for 3 copper wire technologies.

The experimental results show that over 1 kHz, standard round wire cannot be used. To limit copper losses, the transformer coils must be made with Litz wires made of enameled strands as small as possible [48,49].

## 4. Discussion

The specific core loss measured at medium frequency in a GOES wound core (Section 2.2) is higher than for other magnetic materials [37]. This result seems to be a disadvantage for the GOES wound cores. For giving a good understanding of the interest of the proposed solution, it is necessary to situate the transformer in the context of high-power SST cells.

### 4.1. Pre-Design of a High-Power SST Cell

Let us consider a pre-design of a cell of the modular SST presented in Figure 1, which is 7 MW. This DC−DC SST heart is placed between to AC−DC standard bidirectional converters and used for connecting a 20 kV AC line to the 400 V AC grid. It is made of 5 identical cells of 1.4 MW. The DC input voltage of a cell is $U_A = \frac{20\sqrt{2}}{5} = 5.65$ kV for a

DC output voltage that can be tuned up to 650 V in order to allow a tuning marge. The transformer must be designed starting from the apparent power $S$ that depends on the active power $P$ and the power factor $PF$.

$$S = \frac{P}{PF} \tag{13}$$

The power factor $PF$ depends on the detailed design of the transformer (magnetizing current, leakage inductance) and on the DAB commands (pulse width, phase shift). The transformer pre-design is made considering $FP = 0.7$ that corresponds to $S = 2\ MVA$.

The apparent power $S$ of a transformer depends on its global size expressed by the winding area $A_W$ for coils and on core area $A_C$ for the magnetic flux. The filling factor $K_F$ of the core window, the RMS current density $J_{RMS}$, the maximum value of the flux density $B_m$ and the frequency $f$ play also an important role. The transformer's apparent power is given by Equation (14) defined for square voltages. This expression can also be used for sine voltages when coefficient 4 is changed to 4.44 [50].

$$S = 4\ K_F\ A_W\ A_C\ J_{RMS}\ B_m\ f \tag{14}$$

The transformer design must be made for a given apparent power $S$, its size can be estimated using the product $A_W\ A_C$ Equation (15).

$$A_W\ A_C = \frac{S}{4\ K_F\ J_{RMS}\ B_m\ f} \tag{15}$$

The secondary voltage $U_B$ corresponds to the secondary turn number $N_2$ Equation (16) and the voltage ratio yields the primary turn number $N_1$ Equation (17).

$$N_2 = \frac{U_B}{4\ A_c\ B_m\ f} \tag{16}$$

$$N_1 = N_2\ \frac{U_A}{U_B} \tag{17}$$

Several choices must be made for defining the GOES wound core dimensions, which are detailed in Figure 17.

The core is made of $N_s$ GOES strips of width $b$ and thickness $a$; therefore $N_S a$ and $b$ define the core area for the flux $A_c = N_S a\ b$. The window area is $A_w = b\ c$. In this figure, the red dotted lines show the average flux line and the green dotted line the average current line that correspond to the average turn length $l_T$. Equation (18) gives the average flux line $l_C$ and Equation (19) the average turn length $l_T$.

$$l_C = 2c + 2d + \pi\ \frac{N_S\ a}{2} \tag{18}$$

$$l_T = \pi\ (N_S\ a + d) \tag{19}$$

Noting $\rho_{Fe}$ the density of the GOES and $\rho_{Cu}$ the copper one, the core mass $m_C$ and copper one $m_{Cu}$ are expressed by Equations (20) and (21).

$$m_C = \rho_C\ l_C\ A_C \tag{20}$$

$$m_{Cu} = \rho_{Cu}\ l_T\ K_F A_W \tag{21}$$

The specific core losses $P_{CS}$ measured in Section 2.2 and the core mass yield the core losses.

$$P_C = P_{CS}\ m_C \tag{22}$$

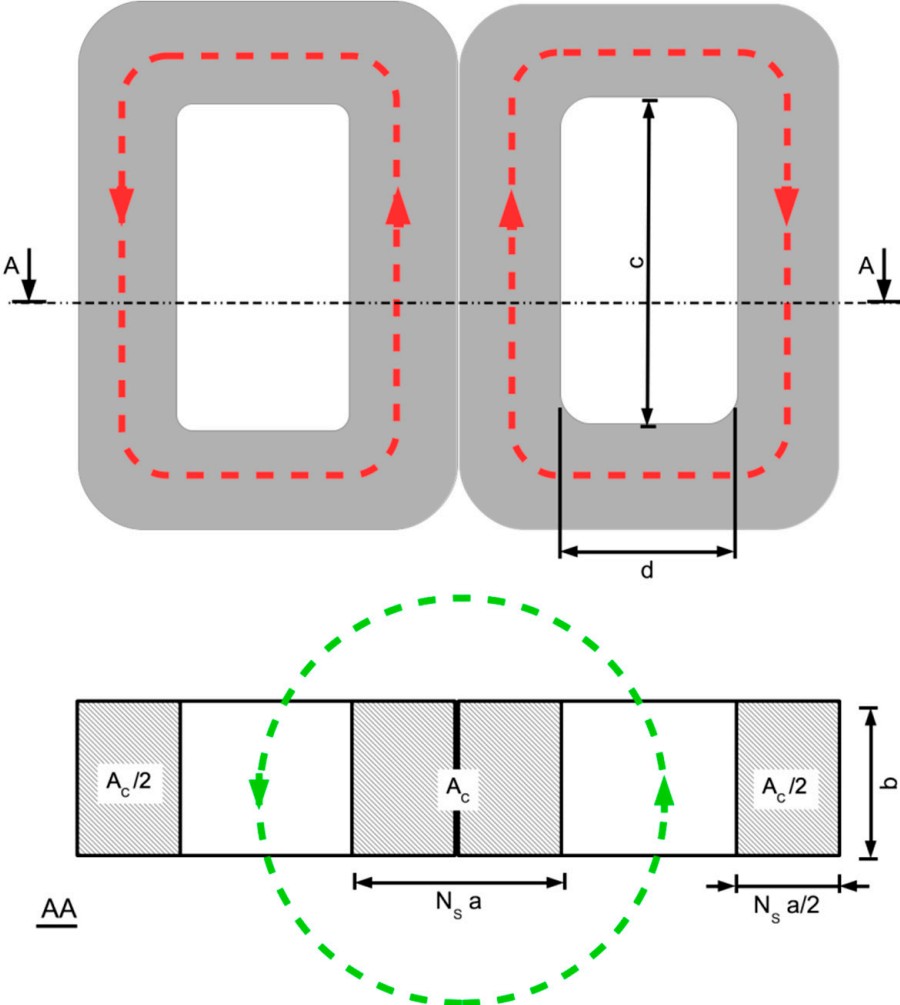

**Figure 17.** GOES wound core functional dimensions needed for the pre-design of the medium frequency transformer of 2 *MVA*–2 kHz. The dotted lines are the average field lines in the core and the average current line in the winding.

The copper losses are computed from the copper conductivity at the winding operating temperature $\sigma_{Cu}$:

$$P_{Cu} = \frac{1}{\sigma_{Cu}} K_F A_W J_{RMS}^2 l_T \tag{23}$$

For the considered transmitted active power $P$, the transformer efficiency is given by:

$$\eta = \frac{P}{P + P_C + P_J} \tag{24}$$

The shorter average current line (average turn length), for a given core area $A_C$, is obtained for a square central leg.

$$b = N_S \, a \tag{25}$$

The ratio $c/d$ is chosen considering the know-how of the wound core manufacturers [51]. For this pre-design, the choice among the data of this standard is $c = 2d$ (rectangular core window) easy to supply.

The filling factor $K_F$ must be low because of the large volume of electrical insulation system for high voltages [52] and the need of cooling airflows as explained in Section 3.1. The choice is $K_F$= 25%. The other choices are $J_{RMS}$ = 5 A/mm$^2$, $B_m$ = 1.2 T and $f$ = 2 kHz. First, Equation (15) is used for determining the product $A_W \, A_C$ then,

the other dimensions are calculated from $(N_S\ a)$, considered as the input parameter for calculus. With $Ac = (N_S c)^2$, Equation (16) yields $N_2$ that must be round to an integer number; Equation (17) gives $N_1$. Knowing $A_c$, $c$ and $d$ depends on $A_W$ given by Equation (15) and the choice $c = 2d$. The copper and c masses $m_C$ and $m_{CU}$ are given by Equations (21) and (20). The total mass is $m = m_{CU} + m_C$. The copper losses are computed with Equation (23) considering the copper conductivity at 150 °C. The core loss comes from Equation (22) and an experimental point of Figure 7 at 200 °C for 1.3 T–2 kHz (105 W/kg). The efficiency $\eta$ for each $(N_S\ a)$ value comes from Equation (24); the last value is the transformer specific power $S_M = S/m$. Full results are given in Table 1.

**Table 1.** Functional sizes and performances of the transformer computed from the side of the square central core square core $N_s a$. The columns are: number of turns ($N_2$, $N_1$); core dimensions (b, c, d); copper and core and total masses ($m_{Cu}$, $m_C$, $m$); copper and core  losses ($P_{Cu}$, $P_C$ ); efficiency ($\eta$ ) and specific power ($S_M$). The conditions are: $S = 2\ MVA$, $f = 2$ kHz, $J_{RMS} = 5$ A/mm$^2$, $B_m = 1.3$ T and a window filling factor of 25%.

|   | $N_s a$ | $N_2$ | $N_1$ | $b$ | $c$ | $d$ | $m_{Cu}$ | $m_C$ | $m$ | $P_{Cu}$ | $P_C$ | $\eta$ | $S_M$ |
|---|---|---|---|---|---|---|---|---|---|---|---|---|---|
|   | (mm) | | | (mm) | (mm) | (mm) | (kg) | (kg) | (kg) | (kW) | (kW) | (%) | (kVA/kg) |
| 1 | 50 | 24 | 208 | 50 | 351 | 175.5 | 173.6 | 28.3 | 201.9 | 12.6 | 3 | 98.90 | 9.90 |
| 2 | 60 | 16 | 137 | 60 | 292 | 146 | 106 | 34.8 | 148.8 | 7.7 | 3.7 | 99.20 | 14.28 |
| 3 | 70 | 12 | 104 | 70 | 251 | 185.5 | 70.8 | 41.7 | 112.5 | 5.1 | 4.4 | 99.33 | 17.77 |
| 4 | 80 | 10 | 78 | 80 | 219 | 109.5 | 50.6 | 49.1 | 99.7 | 3.7 | 5.2 | 99.37 | 20.05 |
| 5 | 90 | 8 | 70 | 90 | 195 | 97.5 | 38.1 | 57.1 | 95.1 | 2.8 | 6 | 99.38 | 21.02 |
| 6 | 100 | 6 | 52 | 100 | 175 | 87.5 | 29.8 | 65.7 | 95.5 | 2.2 | 6.9 | 99.36 | 20.94 |

Table 1 shows that the best functional dimensions are given by line 5 for these hypotheses. This pre-design shows that the high specific core losses are not really a problem for the efficiency because the high-power transformer core mass is relatively low. The efficiency is over 99.3% for a specific power in the range of 20 kVA/kg.

*4.2. On-Going Work*

The pre-design described in the previous section is based on several rather pessimistic hypotheses that need to be refined by a thermal study. The relatively low current density, $J_{RMS} = 5$ A/mm$^2$ can undoubtedly be increased because of the cooling airflows imposed by the centrifugal fans. These airgfows has also a positive influence on the winding temperature. This aspect of the hot core transformer cooling is under study with experimental investigations on a GOES wound core of similar size (Table 1—line 5).

The low feeling factor $K_F = 25\%$ is based on a previous study on the common-mode insulation of a smaller high-voltage SST cell [52]. Proportionally, a larger transformer offers more room for the electrical insulation system. A more detailed study will certainly give a larger filling factor.

Experiments at 2 kHz show that the acoustic pollution of high-power SSTs must be considered. The experience gained during the tests proves that, at 2 kHz, the SST is very noisy. It must be placed in an electrical cabinet reinforced with thick acoustic insulation panels. At higher frequencies, over 4 kHz, the noise is only an unpleasant high-pitched whistle that should be easier to attenuate. The specific power will also increase. The higher core losses will demand a reduction of the flux density. A deeper study of the balance between the maximum flux density $B_m$ the operating frequency $f$, the specific power $S_M$ the maximum temperature in the winding and the noise level of the medium-frequency high-power transformer is necessary. A transformer designed considering the results of Table 1—line 5, is under construction; it will be the central element of the next experimental study.

Noise pollution is a complex problem that needs a specific theoretical approach. Reference [29] that presents a 20 kVA–3 kHz transformer made with an amorphous core, also denotes that the transformer becomes very noisy over 0.3 T. The studies [53] and [54] presents investigations of acoustic noise sources in medium frequency transformers are interesting theoretical starting points.

Considering the manufacturer's point of view, it is difficult and expensive to buy small quantities of specific Litz wires. We will also work on this problem by proposing cheaper aluminum foil windings.

### 5. Conclusions

The paper presents theoretical and experimental investigations on GOES wound core transformers designed for operating at medium frequencies (2–5 kHz). The proposed design uses the good performances of this soft magnetic material at high temperatures. It offers a specific mechanical structure that includes a thermal insulation between the core and the winding in order to take the benefit from the reduction of the core losses at high core temperature. The GOES is a low-cost soft magnetic material with a very stable crystalline structure. It has the benefits of long industrial experience proving the very long-life spans of this material. Compared to the SST technologies currently used, the proposed structure opens the way to the design of higher power compact SST cells operating at medium frequencies, with lower switching losses in converters. The next step is to perform an experimental study of the cooling structure for several $flux\ density \times frequency$ products in order to propose a full design procedure that minimizes the total mass of the transformer. The further step will be the acoustic study at frequencies defined by the full design procedure of high-voltage high-power SST cells at a moderate cost.

**Author Contributions:** Conceptualization, D.R. and E.N.; methodology, D.R.; software, K.K.; validation, D.R., K.K. and E.N.; formal analysis, D.R; investigation, D.R.; resources, D.R. and K.K.; writing—original draft preparation, D.R.; writing—review and editing, E.N. and P.N; visualization, P.N.; supervision, D.R.; project administration, D.R. All authors have read and agreed to the published version of the manuscript.

**Funding:** This research received no external funding.

**Institutional Review Board Statement:** Not applicable.

**Informed Consent Statement:** Not applicable.

**Conflicts of Interest:** The authors declare no conflict of interest.

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
