# Peer review of "Design of High-Power Solid-State Transformers with Grain-Oriented Electrical Steel Cores"

_electronics, doi:10.3390/electronics11152398_

Round 1

Reviewer 1 Report

The work is very interesting and seems to be an original. The work well done but have some  corrections needed to improve the quality of the presentation. I enlist them below:

1.     The Abstract should contain answers to the following questions: What problem was studied and why is it important? What methods were used? What are the important results? What conclusions can be drawn from the results? What is the novelty of the work and where does it go beyond previous efforts in the literature? Please include specific and quantitative results in your Abstract, while ensuring that it is suitable for a broad audience.

2. "Introduction" section is unbalanced. In the other word, there is a time gap between the novel papers related the year 2020, and the other reviewed papers. The papers published in the current year and the last three years should be added.  Also, the literature review is not comprehensive given the topic.

3.     Captions for figures and tables should be checked again. There is a significant lack of information. Please provide readers enough information on them.

4.     It is helpful to complete the description of how to collect data, data processing scenarios, and interpret the data collection.

5.     The conclusion must answer whether the proposed method can solve the research problem and achieve the objective. How can the numerical approach answer the existing issues? What is the most important result? What are the implications for science and technology development?

6.     Outlook and future perspectives can be included in the conclusion part.

7.     The discussion seems inadequate, and this is too short for a reputable international journal. Many graphical presentations are similar; it is worth thinking about expressing with other graphics (if possible).

8.The authors should emphasize and explain the current study's novelty, which differs remarkably from previous research

Author Response

Response to Reviewer 1 Comments

Thank you very much for your sound analysis. It has been very useful for improving our paper. All your comments have been considered. The English language has been checked by a Scottish colleague. Please find below our detailed answers.

Please, find also attached the revised version of our paper.

The work is very interesting and seems to be an original. The work well done but have some corrections needed to improve the quality of the presentation. I enlist them below:

  1. The Abstract should contain answers to the following questions: What problem was studied and why is it important? What methods were used? What are the important results? What conclusions can be drawn from the results? What is the novelty of the work and where does it go beyond previous efforts in the literature? Please include specific and quantitative results in your Abstract, while ensuring that it is suitable for a broad audience.

The abstract has been deeply changed. The problem is better situated in the context of deep changes of the electric grid due to the societal demand tostrongly reduce greenhouse gas emissions. The main novelty of the paper is the demonstration that core losses are lower for rectangular voltages than for sine ones. This experimental constatation and its theoretical explanation open the way to use GOES wound cores for building SST cells over 1MW at a reasonable cost.

  1. "Introduction" section is unbalanced. In the other word, there is a time gap between the novel papers related the year 2020, and the other reviewed papers. The papers published in the current year and the last three years should be added. Also, the literature review is not comprehensive given the topic.

The bibliography is deeply changed with more references to recent works chosen among many publications on SSTs and magnetic materials able to operate at medium frequencies. In my opinion, it is not possible to provide an exclusive review of previous works for such many possible references.

  1. Captions for figures and tables should be checked again. There is a significant lack of information. Please provide readers enough information on them.

Details were added in captions for providing to the reader synthetic data on each figure.

  1. It is helpful to complete the description of how to collect data, data processing scenarios, and interpret the data collection.

Details were added to data collected from measurements. The text added in section 2.2 is short because we used classical measurement methods.

  1. The conclusion must answer whether the proposed method can solve the research problem and achieve the objective. How can the numerical approach answer the existing issues? What is the most important result? What are the implications for science and technology development?

The conclusion remains short because it follows the enlarge discussion that details these points.

  1. Outlook and future perspectives can be included in the conclusion part.

The authors choose to add a part "on-going works" in the section dedicated to the discussions for describing the research we will perform next year. In our opinion, this presentation better shows the concrete aspects of the perspectives.

  1. The discussion seems inadequate, and this is too short for a reputable international journal. Many graphical presentations are similar; it is worth thinking about expressing with other graphics (if possible).

We agree with you, this section has been totally rewritten and based on the pre-design of a 2 MVA transformer operating at 2 kHz.

  1. The authors should emphasize and explain the current study's novelty, which differs remarkably from previous research

The main novelty is to design a transformer with a GOES wound core much hotter than coils for taking the benefits of lower core losses with similar copper losses. This choice provides a higher compacity and opens the way to larger power per SST cells. This was highlighted in the introduction.

Reviewer 2 Report

According to the Authors, "The paper proposes a new concept for designing medium-frequency high-power transformers (over 500 kVA) for SST cells. The concept uses the good properties of GOES thin strips at high temperatures. With these transformers and standard 6500 V insulated gate bipolar transistors (IGBTs) modules [27], it is possible to design very large power 78 SSTs (several MW) at a reasonable cost, with a limited number of cells [28]. The structure proposed uses wound cores made of thin GOES strips. In such cores the magnetic field is in the strip rolling direction at any point, which corresponds to the easy magnetisation axis. This direction corresponds to the higher magnetic permeability and saturation flux  density. The high permeability allows a core design with long strips that offer a large core window for the winding. ... The end of the paper gives key elements for designing a high-power medium-frequency transformer with a core at a much higher temperature than for coils in order to increase the transformer compacity. " The text is well-written, structured, referenced and interesting to the readers. However, just a few improvements listed in the following could be made in order for the paper to be accepted.

In page 4, there is an error in "the relative value of conduction losses decreases as U is increase(s)."

In the text, write "Figure" with capital "F".

The major problem with the text lies in the discussion. It lacks references to the literature. For instance, in "this limit is in the kHz range for the available power semiconductor technologies."  and "An operating frequency in the kHz range requires to take care of the acoustic pollution of high-power SSTs. The experience gained during the tests proves that, at 2 kHz, the SSTs must be placed in an electrical cabinet reinforced with acoustic insulation panels. Over 4 kHz, the noise is only a high-pitched whistle that does not disturb the standard environment of high-power electrical equipment." Bringing references in order to compare the obtained performance would be desirable too.

Author Response

Response to Reviewer 2 Comments

Thank you very much for your sound analysis, which has been very useful for improving our paper. All your comments have been considered; the English language has been checked by our Scottish colleague.

Please, find below our detailed answers and attached the revised version of our paper.

According to the Authors, "The paper proposes a new concept for designing medium- frequency high-power transformers (over 500 kVA) for SST cells. The concept uses the good properties of GOES thin strips at high temperatures. With these transformers and standard 6500 V insulated gate bipolar transistors (IGBTs) modules [27], it is possible to design very large power SSTs (several MW) at a reasonable cost, with a limited number of cells [28]. The structure proposed uses wound cores made of thin GOES strips. In such cores the magnetic field is in the strip rolling direction at any point, which corresponds to the easy magnetisation axis. This direction corresponds to the higher magnetic permeability and saturation flux density. The high permeability allows a core design with long strips that offer a large core window for the winding. ... The end of the paper gives key elements for designing a high-power medium-frequency transformer with a core at a much higher temperature than for coils in order to increase the transformer compacity. " The text is well-written, structured, referenced and interesting to the readers.

However, just a few improvements listed in the following could be made in order for the paper to be accepted.

In page 4, there is an error in "the relative value of conduction losses decreases as U is increase(s)."

The text is corrected

In the text, write "Figure" with capital "F".

The whole text has been checked for applying this sound advice.

The major problem with the text lies in the discussion. It lacks references to the literature. For instance, in "this limit is in the kHz range for the available power semiconductor technologies." and "An operating frequency in the kHz range requires to take care of the acoustic pollution of high-power SSTs. The experience gained during the tests proves that, at 2 kHz, the SSTs must be placed in an electrical cabinet reinforced with acoustic insulation panels. Over 4 kHz, the noise is only a high-pitched whistle that does not disturb the standard environment of high-power electrical equipment." Bringing references in order to compare the obtained performance would be desirable too.

We agree with these comments. Several references to previous works were added. We also agree that the aspect relative to acoustic pollution was not included in the paper. We don’t made any noise measurements at this level of the development of the “hot GOES wound core transforer” new concept. Noise pollution is a imoprtant lock that must be overcome. We change totally the section 4 “discussion” in order to discribe more accurately the following steps of our study. We are now at the level “proof of convept”. The paper proves that the concept of a hot GOES wound core in coils at standard temperatures can he used for designing hign-power SST cells. The next step will be to propose a detailed design bease of a prototype of actual sizes for testing the thermal balance at several values of the procuct Bm*f. The optimisarion of this procuct is directly linked to the cooling sytem. Our choice is to make this optimisation using experimental investigations. The accoustic pollution of the medium frequency transformer depens strongly of this procuct. We chose to overcome this important lock in a second step, with several acceptable values of the procuct Bm*f that corresponds to the functional point of view (high power densities and efficiencies, temperature security margins at the hottest points of coils). If the actual operating frequency is higher, the acoustic problem will be easier to solve.

Round 2

Reviewer 1 Report

The authors have addressed all comments and the article can be accepted in its present form.